# Recent Synthesis Developments of Organoboron Compounds via Metal-Free Catalytic Borylation of Alkynes and Alkenes

**DOI:** 10.3390/molecules24010101

**Published:** 2018-12-28

**Authors:** Yanmei Wen, Chunmei Deng, Jianying Xie, Xinhuang Kang

**Affiliations:** Faculty of Chemistry and Environmental Science, Guangdong Ocean University, Zhanjiang 524088, China; wenym@gdou.edu.cn (Y.W.); dcm2382405@163.com (C.D.); xiejy1961@126.com (J.X.)

**Keywords:** organoboron compound, metal-free catalytic boration, nucleophilic boron, alkyne, alkene

## Abstract

Diboron reagents have been traditionally regarded as “Lewis acids”, which can react with simple Lewis base to create a significant nucleophilic character in one of boryl moieties. In particular, bis(pinacolato)diboron (B_2_pin_2_) reacts with simple Lewis bases, such as *N*-heterocyclic carbenes (NHCs), phosphines and alkoxides. This review focuses on the application of trivalent nucleophilic boryl synthon in the selective preparation of organoboron compounds, mainly through metal-free catalytic diboration and the β-boration reactions of alkynes and alkenes.

## 1. Introduction

Organoboron compounds play an important role in organic chemistry as they can be converted into a wide variety of functional groups [1,2,3,4,5,6,7,8,9,10,11,12,13]. Thus, the synthesis of different types of organoboron compounds continues to be an important research field [14,15,16,17,18,19,20]. Conventional methods for their preparation are based on either reactive organometallic reagents or transition-metal-mediated processes [21,22,23,24,25,26,27,28,29,30,31,32,33,34,35,36,37]. Recently, great effort has been made towards the development of transition-metal-free methods with high chemoselectivity or regioselectivity and environmental sustainability [38,39,40,41,42,43,44,45,46,47,48]. The transition-metal-free catalyzed borylation reaction of alkynes and alkenes offers an attractive method for generating valuable organoboron compounds. The interaction of a diboron reagent, in particular bis(pinacolato)diboron (B_2_pin_2_), with a simple Lewis base, such as *N*-heterocyclic carbenes (NHCs), phosphines and alkoxides, generates Lewis acid-base adducts, in which the formally intact sp^2^ boryl moiety becomes the important source of nucleophilic boryl species, resulting in an appropriate approache towards transition-metal-free selective preparation of organoboron compounds. This review focuses on the application of trivalent nucleophilic boryl synthon in the selective preparation of organoboron compounds, mainly through diboration and β-boration reactions of alkynes and alkenes.

## 2. Organoboron Compounds via Transition-Metal-Free Catalytic Diboration of Unsaturated Hydrocarbons

In recent years, the preparation of 1,2-diborated or 1,1-diborated products has rapidly grown due to the addition of diboryl compounds to unsaturated organic compounds in the transition metal-free catalysis. The first diboration reaction of non-activated alkenes in the absence of transition metal complexes was reported by Fernández and coworkers (Scheme 1a) [49]. The diboration of non-activated olefins, such as terminal and internal alkenes, allenes and vinylarenes, can be produced via the formation of a 1,2-diborated product in a tetrahydrofuran (THF) solvent at 70 °C for ~6–16 h. The interaction of B_2_pin_2_ (**1**) and a methoxide anion, derived from the combination of Cs_2_CO_3_ and MeOH, can generate in situ the adduct [B_2_pin_2_OMe]^−^. This adduct reacts with the unsubstituted carbon atom (C^1^) of the C=C double bond via the transition state (TS1), which is the overlap between the strongly polarized B-B σ orbital of the activated diboron reagent and the C–C π* orbital of the olefin. When the B–B bond weakens, the negative charge density on the C^2^ of the C=C double bond increases. The negatively charged olefin-boryl olefin-B(pin) fragment attacks the boron atom with an electrophilic character to form the second transition state (TS2), which then protonates to form the diborated product (Scheme 2). The adduct [B_2_pin_2_OMe]^−^ has been fully characterized by both ^11^B-NMR spectroscopy and ESI-MS. A later report extended the methodology using unsymmetrical Bpin–Bdan (dan represents 1,8-diaminonaphthalene) (**2**) as the diboron reagents, where the methoxide selectively interacted with the stronger Lewis acid Bpin moiety. The Bdan moiety appears in the internal position (Scheme 1b) [50].

In 2015, Sawamura reported the formation of α,β-diboryl acrylates through trialkylphosphine catalyzed anti-selective diboration of the C≡C triple bond (Scheme 3a) [51]. Through cross-coupling reaction of the differentiated heteroatom substituents of the α,β-diboryl acrylates in a stepwise manner, a diverse array of unsymmetrical tetrasubstituted alkenes can be accessed. Very recently, Santos employed an unsymmetrical Bpin–Bdan **2** as a diboron reagent for the direct diboration of alkynamides, which display broad functional group tolerance. The transition-metal-free catalyzed diboration of alkynamides with Bpin–Bdan, where the Bdan and Bpin moiety install α- and β-carbon atoms respectively. (Scheme 3b) [52].

In 2014, Uchiyama and coworkers reported that the diboration of propargylic alcohols with diboron reagents in the presence of a stoichiometric amount of n-BuLi enables the trans-diboration of alkynes through aqueous workup. The present trans-diborylation gave oxaborole moieties which are key structural platforms and potent pharmacophores in material and pharmaceutical sciences (Scheme 4) [53]. Very recently, the direct diboration of alkynes using a more reactive unsymmetrical pinB-Bmes_2_ (**3**) as the diboron reagent was reported by Yamashita’s group. The mixture of two cis-isomers and one trans-isomer was obtained when the base-catalyzed diboration reaction of the terminal alkynes was conducted in the system consisting of n-BuLi and 1,2-dimethoxyethane in toluene (Scheme 5) [54].

In 2015, Sawamura revealed a new method for the synthesis of 1,1-diborylalkenes through a Brönsted base catalyzed reaction between bis(pinacolato)diboron and various terminal alkynes containing propiolates, propiolamides and 2-ethynylazoles (Scheme 6a) [55]. In 2018, the Brönsted base catalyzed diboration of strong electron deficient alkynes, including various propiolates, ynones and propiolamides, was developed by Song’s group (Scheme 6b) [56]. The terminal and internal alkynes were allowed to react under the “optimal” conditions, and good to excellent yields of the corresponding gem-diborylalkanes were obtained.

The asymmetric organocatalytic 1,2-diboration of alkenes has been accomplished utilizing economically accessible chiral alcohols as additives to form the Lewis acid-base ^*^RO^-^→bis(pinacolato)diboron adduct that is added to cyclic and non-cyclic alkenes, providing moderate enantioselectivity. The authors reported moderate conversions and moderate levels of enantioselectivity for the synthesis of the 1,2-diborated product (Scheme 7a) [57]. Alternatively, the enantioselective diboration of alkenes can be achieved using inexpensive chiral carbohydrates, such as the pseudoenantiomeric glycol 6-tertbutyldimethylsilyl-1,2-dihydroglucal (**4**) and dihydrorhamnal (**5**) (Scheme 7b) [58]. The alkoxide-catalyzed directed diboration of alkenyl alcohols through the intramolecular activation of the diboron reagent and the selective delivery of the nucleophilic boryl unit offers access to the diboration of cyclic and acyclic homoallylic and bishomoallylic alcohol substrates (Scheme 7c) [59].

## 3. Organoboron Compounds via Transition-Metal-Free Catalytic β-boration of α,β-Unsaturated Compounds

### 3.1. N-Heterocyclic Carbene Catalysis

In 2009, Hoveyda and coworkers firstly reported that an imidazolium salt **6** in the absence of a transition metal was able to promote the boron conjugation addition of cyclic and acyclic α,β-unsaturated ketones and esters, providing β-boryl carbonyls (Scheme 8) [60]. It is noteworthy that in the presence of a base (NaOtBu) only, the borylation reaction did not take place. It was proposed that in this process a nucleophilic *N*-heterocyclic carbene (NHC) could associate with the diboron reagent in presence of a base and an in situ generated NHC·B_2_(pin)_2_ complex. The diboron reagent becomes nucleophilic and is able to transfer the “intact” sp^2^ boryl group to the activated olefins in boron conjugate additions. A proton from alcohol additives or aqueous workup served as the electrophilic counterpart of the boron nucleophile (Scheme 9). Density functional theory (DFT) calculations and ^11^B-NMR studies were carried out to determine the exact structure of the proposed sp^2^–sp^3^ hybridized NHC·B_2_(pin)_2_ adduct. The DFT calculations showed polarization of the B–B bond and diminished electrophilic character of the two boron atoms, where the electron density was strongly polarized towards the unbound boron center. However, due to the weak association of the NHC to the sp^2^–sp^2^ diboron, causing dynamic exchange, the ^11^B-NMR signal for B_2_pin_2_ disappeared within five minutes after treatment with the NHC. In 2012, the crystal structure of the neutral NHC adduct **7** was reported by Marder and coworkers (Scheme 10) [61].

In a subsequent study, Hoveyda reported further work on enantioselective method for boron conjugate addition to α,β-unsaturated carbonyls with a chiral NHC **8** in combination with 1,8-diazabicyclo[5.4.0]undec-7-ene (DBU). This method can tolerate a broad range of substrates, including acyclic and cyclic α,β-unsaturated ketones, as well as acyclic esters, amides, and aldehydes. The desired β-boryl carbonyls are obtained in up to a 94% isolated yield and an >98:2 enantiomer ratio (Scheme 11a) [62]. A later report extended this methodology to β-disubstituted enones, generating products containing boron-substituted quaternary carbon with enantioselectivity (Scheme 11b) [63]. The use of diboranes and *N*-heterocyclic carbenes in the absence of a transition metal complex has been reported by other groups [64,65].

### 3.2. Base Catalysis

In 2012, Fernández reported the utility of the base-catalyzed method for the β-boration of acyclic and cyclic activated olefins in cases involving α,β-unsaturated amides and phosphonates [66]. The methodology could be applied to different diboron reagents, such as bis(pinacolato)diboron (B_2_pin_2_), bis(catecholato)borane (B_2_cat_2_) and so on. Alternative inorganic and organic bases were explored, such as NaOMe, LiOMe, NaOtBu, K_2_CO_3_, CsF and Verkade’s base, the latter being the most efficient (Scheme 12). With an excess of MeOH, Verkade’s base is completely protonated and the generated MeO^−^ forms a Lewis acid-base adduct with B_2_pin_2_. Computational studies identified that there was an overlap between the strongly polarized B–B σ orbital and the C–C π* orbital, and DFT calculations revealed that in adducts the sp^2^ boron atom gains a strong nucleophilic character because the sp^3^ boron atom loses negative charge density upon the charge transfer from the Lewis base. The intact sp^2^ boron atom corresponds to a nucleophilic attack at the β-carbon atom of the activated olefins, forming a transition sate (TS), which releases the “(pin)BOMe” and leads to the formation of an anionic intermediate, which is then protonated to provide the β-boration product (Scheme 13).

A later report extended the methodology to non-symmetrical diboron reagents, such as Bpin–Bdan **2**, where the RO^−^ selectively interacted with the stronger Lewis acid Bpin moiety (Scheme 14a) [67]. Experimental and theoretical investigation confirmed that the interaction of RO^−^ and the Bpin moiety preferred to form a [RO→Bpin–Bdan]^−^ intermediate, which then resulted in exclusively the β-carbon Bdan carbonyl compound with high yields. The adduct [MeO→Bpin–Bpin]^−^ which efficiently mediates the β-boration of α,β-unsaturated imines formed in situ was reported by Fernández’s group (Scheme 14b) [68]. From a theoretical point of view, the mechanism of β-boration of α,β-unsaturated imines has been thought to involve quaternization of the diboron reagent with methoxide. The function of the phosphine has been regarded to the ion pair formation.

### 3.3. Phosphine Catalysis

In 2010, Fernández reported the β-borylation of α,β-unsaturated compounds using a trivalent phosphorus nucleophile in combination with catalytic amounts of bases and stoichiometric amounts of methanol (Scheme 15) [69]. Further screening showed that a variety of phosphorus compounds including achiral, chiral mono and bidentate phosphines could also give reasonable conversions without a transition metal. The process was proposed that the phosphine interacted with the empty orbital of one of the boron atoms to result in the heterolytic cleavage of the B–B bond of B_2_pin_2_. Upon such interaction, the other boron moiety could act as a nucleophile towards the α,β-unsaturated esters and ketones (Scheme 16) [69].

In a further publication, Fernández presented novel reaction conditions for the β-boration reaction of α,β-unsaturated carbonyl compounds without a transition metal. Eluding the use of Brönsted bases to promote the catalytic active species, the reaction can be carried out by the unique presence of catalytic amounts of phosphine in MeOH (Scheme 17) [70]. The experimental and theoretical studies demonstrated interactions between the phosphine and the substrate. Initially, the phosphine interacts with the substrate to form a strongly basic zwitterionic phosphonium enolate species, which would aid deprotonation of the MeOH to eventually form the ion pair [α-(H),β-(PR_3_)-ketone]^+^[B_2_pin_2_·MeO]^−^ (Scheme 18).

Córdova developed the first metal-free one-pot three-component catalytic selective reaction between a diboron reagent, α,β-unsaturated aldehydes, and 2-(triphenylphosphoranylidene) acetate esters for the synthesis of homoallylboronates. The multicomponent reactions undergoes a catalytic β-boration of α,β-unsaturated aldehydes to give the β-borylaldehyde. A subsequent Wittig reaction between the β-borylaldehyde and the 2-(triphenylphosphoranylidene)acetate esters yields the corresponding homoallylboronates with high chemoselectivity and regioselectivity (Scheme 19) [71].

## 4. Organoboron Compounds via Transition-Metal-Free Catalytic Borylation of Allyic and Propargylic Alcohols

The base-catalyzed borylation of tertiary allylic alcohols can be efficiently performed to prepare 1,1-disubstituted allyl boronates with a high yields (Scheme 20a) [72]. The reactions are performed in the presence of a catalytic amount of Cs_2_CO_3_ and MeOH to promote the formation of the Lewis acid-base adduct, [Hbase]^+^[MeOB_2_pin_2_]^−^, which may also be responsible for the tandem allylic borylation/diboration of allylic alcohols that generate 1,2,3-polyborated products (Scheme 20a). To complement the study, the scope of the metal-free allylic borylation was further expanded to 1-propargylic cyclohexanol, to form the corresponding product with a working hydroborated triple bond at 50 °C. Interestingly, by increasing the temperature to 90 °C, the alkenyl borane was selectively formed (Scheme 20b). The author suggested a possible mechanism to account for this process. First, the adduct [B_2_pin_2_·MeO]^−^ was formed through a base-mediated nucleophilic attack at one of the Bpin moieties of B_2_pin_2_. Second, the other Bpin unit with nucleophilic activity attacks the terminal carbon atom of the allylic alcohol, which leads cleavage of the C–O bond. Last, the base catalyst is regenerated via the reaction of the conjugated acid of the base (BaseH^+^) and the OH group (Scheme 21).

Another facile synthetic process for converting tertiary allylic alcohols to corresponding allylic boronates in moderate to good yields was reported. The mechanistic approach suggested that allylic boronate formation was achieved via the sequential processes of Fernández-type alkoxide-mediated diborylation of an alkene and the boron-Wittig reaction (Scheme 22) [73].

On the basis of “pseudo-intramolecular activation” [53] of diborons enabling the trans-selective diboration of alkynes, Uchiyama and coworkers found that the carboboration reaction was enabled by pseudo-intramolecular activation of alkynylboronates using propargylic alcohols to produce oxaborole products (Scheme 23) [74]. The synthetic utility of this product was investigated using medicinal chemistry and strong blue fluorescence emission.

## 5. Organoboron Compounds via Transition-Metal-Free Catalytic Hydroboration of Unsaturated Hydrocarbons

In 2016, Song reported that the transition-metal-free catalytic borylation of arylacetylenes or vinyl arenes can been efficiently performed to produce alkylboronates through tandem borylation and protodeboronation (Scheme 24a) [75]. This method has high regioselectivity and chemoselectivity, good to excellent yields and can tolerate a broad range functional groups. The transition-metal-free method for borylation of alkynes and alkenes with bis(pinacolato)diboron can also easily afford vinylboronates (Scheme 24b) [76] or diboryl alkanes (Scheme 24c) [77]. In 2018, Fernández and coworkers reported that the 1,4-hydroboration of 1,3-dienes was followed by an in situ diboration reaction of the internal double bond to produce a 1,2,3-triboration product (Scheme 24d) [78].

A new approach to the transition-metal-free synthesis of alkenyl boronates was based on the hydroboration of the distal double bond of allenamides (Scheme 25) [79]. In order to obtain exclusively the complete stereoselective product, the acyl groups on the amine moiety were crucial, which formed a stable allylic anion intermediate that was further regioselectively protonated to obtain a *Z*-isomer.

## 6. Conclusions

In this review, we have presented numerous very useful processes for the synthesis of valuable organoboron compounds, which involve transition-metal-free catalyzed diboration, β-boration and boryl substitution reactions. The key to the success of this reaction is to use a simple Lewis base capable of activating the diboron reagent to create a significant nucleophilic boryl synthon. The reactions tolerate a wide variety of functional groups and progress under metal-free reaction conditions, thus avoiding contaminating the final boron products with heavy metals, a problem which is frequently encountered in the manufacturing process. Although the metal-free catalyzed borylation of alkynes and alkenes is relatively unexplored at this time, there is no doubt that the future is promising for this reaction and there will be many exciting discoveries in this field.

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
