# Peer review of "Recent Synthesis Developments of Organoboron Compounds via Metal-Free Catalytic Borylation of Alkynes and Alkenes"

_molecules, 2018, doi:10.3390/molecules24010101_

Round 1
Reviewer 1 Report
The review described recent progress of borylation of alkynes and alkenes.
The authors have to consider the construction the manuscript.
No compound numbers, no chemical yields, no descriptions for selectivity in the schemes etc caused many readers confused to read the review.
Furthermore, reviewer can not recognized that the authors are specialists in the fields and afraid they type of the review will publish in open access journal.
Author Response
Based on reviewer’s comments, we have made modification on the manuscript. We acknowledge reviewer’s comments and suggestions very much, which are valuable in improving the quality of our manuscript.
The authors have to consider the construction the manuscript.
Have revised(See the revised manuscript).
No compound numbers, no chemical yields, no descriptions for selectivity in the schemes etc caused many readers confused to read the review.
Have inserted compound numbers, chemical yields, and selectivity in the schemes.
Furthermore, reviewer can not recognized that the authors are specialists in the fields and afraid they type of the review will publish in open access journal.
Our research group has been working on synthesis of organoboron compounds. Recently, we
develop one-pot borylation/cross-coupling for the efficient construction of 1,3-dienes. And the investigations on the activity of organoboron compounds is currently ongoing in our laboratory.
Reviewer 2 Report
The presented review article describes the selected examples of methods of the synthesis of various boron-containing organic compounds starting from unsaturated materials like alkenes and alkynes. From the substantive point of view this contribution seems to be correct, the selected reactions are appropriate, however, some editorial mistakes found in the text should be corrected.
1) There is a lack of spaces between text and brackets, especially in the description of schemes (for example, page 1, line 41, page 2, lines 54 and 56, page 3, line 64 and further in the text).
2) Page 1, line 41 - 'protonates' instead of 'protonate'.
3) Page 3, line 75 - 'co-workers' instead of 'co-worker'.
4) Page 4, line 100 - 'N' should be written in italic.
5) Page 5, line 119 - 'dbu' should be written in capitals.
6) Page 5, lines 120 and 121 - 'The desired β-boryl carbonyls isolated yield of 94% and >98:2 enantiomer ratio were obtained' - this phrase is somewhat unclear.
7) Page 5, line 124 - 'N' should be written in italic.
8) Page 9, line 207 - 'co-workers' instead of 'co-worker'.
9) Page 10, line 236 - please check bold in the caption of Scheme 25.
Author Response
Based on reviewer’s comments, we have made modification on the manuscript. We acknowledge reviewer’s comments and suggestions very much, which are valuable in improving the quality of our manuscript.
1) There is a lack of spaces between text and brackets, especially in the description of schemes (for example, page 1, line 41, page 2, lines 54 and 56, page 3, line 64 and further in the text).
Have revised.
2) Page 1, line 41 - 'protonates' instead of 'protonate'.
Have revised.
3) Page 3, line 75 - 'co-workers' instead of 'co-worker'.
Have revised.
4) Page 4, line 100 - 'N' should be written in italic.
Have revised.
5) Page 5, line 119 - 'dbu' should be written in capitals.
Have revised.
6) Page 5, lines 120 and 121 - 'The desired β-boryl carbonyls isolated yield of 94% and >98:2 enantiomer ratio were obtained' - this phrase is somewhat unclear.
Have revised.
7) Page 5, line 124 - 'N' should be written in italic.
Have revised.
8) Page 9, line 207 - 'co-workers' instead of 'co-worker'.
Have revised.
9) Page 10, line 236 - please check bold in the caption of Scheme 25.
Have revised.
Reviewer 3 Report
The manuscript of Wen et al. is a valuable review on synthesis of organoboron compounds from alkenes and alkynes. This can be accepted after minor changes (mistypes).
Row 34 Diboration.
Rows 41, 56, 66 , 79, 83 100, 104, 119132, 133, 135, 149, 157 190, 203, 215 224. Space before brackets e. g. ..product (Scheme 2).
Row 117 In a subsequent study,...
Row 136, 146: increase the size of the superscript "-"
Remark: Section 4. Organoboron Compounds via Transition-metal-free Catalytic other borylation (the title is rather confusing), moreover it should be combined with section 3.
Author Response
Based on reviewer’s comments, we have made modification on the manuscript. We acknowledge reviewer’s comments and suggestions very much, which are valuable in improving the quality of our manuscript.
Row 34 Diboration.
Have revised.
Rows 41, 56, 66 , 79, 83 100, 104, 119132, 133, 135, 149, 157 190, 203, 215 224. Space before brackets e. g. ..product (Scheme 2).
Have revised.
Row 117 In a subsequent study,...
Have revised.
Row 136, 146: increase the size of the superscript "-"
Have revised.
Remark: Section 4. Organoboron Compounds via Transition-metal-free Catalytic other borylation (the title is rather confusing), moreover it should be combined with section 3.
Have revised(See the revised manuscript).
Round 2
Reviewer 1 Report
The order of reference number from lane 316 must be changed.
Did authors check the submitted PDF carefully?
Author Response
We acknowledge the reviewer’s comments and suggestions very much, which are valuable in improving the quality of our manuscript.